# Dietary Factors of *bla*_NDM_ Carriage in Health Community Population: A Cross-Sectional Study

**DOI:** 10.3390/ijerph18115959

**Published:** 2021-06-02

**Authors:** Shuangfang Hu, Ziquan Lv, Qiumei Xiang, Yang Wang, Jianzhong Shen, Yuebin Ke

**Affiliations:** 1Key Laboratory of Molecular Epidemiology of Shenzhen, Shenzhen Center for Disease Control and Prevention, Shenzhen 518055, China; hushuangfang@126.com (S.H.); lvziquan1984@126.com (Z.L.); 13164814279@163.com (Q.X.); 2Beijing Advanced Innovation Center for Food Nutrition and Human Health, College of Veterinary Medicine, China Agricultural University, Beijing 100083, China; vetwangyang@163.com

**Keywords:** *bla*_NDM_, epidemiology, dietary factor, risk analysis, cross-sectional study

## Abstract

Aim: There is an ongoing debate as to what extent antimicrobial resistance (AMR) can be transmitted from dietary to humans via the consumption of food products. We investigated this association between dietary and global spreading carbapenem-resistant gene *bla*_NDM_ Methods: We did a cross-sectional study to assess the risk factors for carrier of *bla*_NDM_ in health community. Healthy adults were recruited from the residents attending Community Healthcare Service in Shenzhen City (Guangdong Province, China), through 1February 2018 to 31December 2019, and 718 pre-participants were included in this study. Questionnaire were obtained and the qualitative food frequency questionnaire (Q-FFQ) were used to assess dietary intake. qPCR was applied to confirm the carrier of *bla*_NDM_ in participants’fecal samples. Multivariable logistic regression was used to estimate the odds ratio (OR) and 95% confidence interval (95% CI) of each outcome according to each dietary factor before and after prosperity score matching (PSM). Results: we showed that a high intake of coarse grain (OR 1.003; 95% CI 1.001–1.005, *p* < 0.01) and root and tuber crops (OR 1.003; 95% CI 1.001–1.004, *p* < 0.05) were independent risk factor for *bla*_NDM_ carrier in health communities, suggesting a possible transfer of AMRbetweendietary andhumans. Surprisingly, we also showed an association between a higher intake of poultry as a protective, which may be explained by the beneficial effects on the gut microbiota. Conclusion: Dietary factors such as intake of coarse grain, root and tuber crops and poultry were associated with *bla*_NDM_ carrier in health communities. The influence of dietary factorson *bla*_NDM_ carrier in the present study provides insights for the tangible dietary advice with guidelines to the routine of people with the risk of *bla*_NDM_ carrier. This demonstrates the role of dietary intake in the prevention of *bla*_NDM_ carrier, since prevention is the best way to control modifiable risk factors. A lower carrier rate of *bla*_NDM_ is helpful to reduce the possibility of transmission and pathogenicity. Further studies on food, microbiota and antimicrobial resistance are necessary to confirm this possible association and unravel underlying mechanisms.

## 1. Introduction

The emergence and spread of carbapenem-resistant *Enterobacteriaceae* (CRE) represents an emergent and serious threat to global public health systems [1]. The main mechanism for CRE is carbapenemases which belong to β-lactamases [2] and divided into three classes as follows: Class A (KPC, SME and so on); Class B (metallo-β-lactamases, including NDM, IMP and VIM); and Class D (OXA-like) [3]. 

Carbapenem antibiotics have never been licensed for veterinary use in any country worldwide. While a growing number of carbapenemases found from non-human sources have been reported [4,5,6,7]. Despite the expansion of the New Delhi metallo-β-lactamase-1 (NDM-1) worldwide, recent years NDM has been frequently found from non-human sources such as food animals, vegetables and water in China [8,9,10,11,12,13,14,15]. Since the food chain has serve as a reservoir for resistance genes, due to the lack of epidemiological data on the role of diet in resistance genes in humans, there is an ongoing debate as to what extent AMR such as carbapenem resistance can be transmitted from dietary intake to humans via the consumption of food products [16]. 

It has been found a distinct susceptibility pattern such as low antimicrobial resistance in community-acquired CRE isolates compared to healthcare-acquired carbapenem-resistant [17]. The epidemiology and risk factors associated with healthcareacquired CRE has been widely studied [18,19]. However, the epidemiology as well as risk factors associated with CRE colonization in health community population remain poorly understood. Resistome *bla*_NDM_ widely exists in many kinds of strains, and could transfer between different strains through plasmids [20]. Therefore, the carrier of *bla*_NDM_ increased the risk of its spread in the population. Considering the frequency which *bla*_NDM_ were found in food stuff and environment, potential associations between the dietary factors and *bla*_NDM_ carrier (among health community population) have not been explored. The primary aim of this studyis to investigate the association between dietary factor and *bla*_NDM_ carrier, using data from a health community population in Shenzhen City, Guangdong Province, China, through1 February 2018 to 31 December 2019.

## 2. Materials and Methods

### 2.1. Study Design and Participants

We did a cross-sectional study to assess the risk factors for carrier of *bla*_NDM_ in health community. Fresh fecal samples were collected from healthy volunteers from 1 February 2018 to 31 December 2019, attending Community Healthcare Service in Shenzhen City, Guangdong Province, China. We compared risk factors for *bla*_NDM_ carrier (*bla*_NDM_-positive cases from communities) with *bla*_NDM_-negative attending from communities. We excluded pregnant women, neonates, volunteers with gastroenteritis and patients with gastrointestinal cancer, gastrointestinal surgery, peptic ulcer, gastrointestinal bleeding, inflammatory bowel disease (e.g., Crohn’s disease, ulcerative colitis), intestinal polyps, intestinal fistula, anal fistula or anal fissure. For the analysis of risk factors of colonization study, we obtained participants’ demographics, ICU admission, residence andproximity to commercial animal farms, education and income, dietary pattern and drinking water and antibiotic use before and during hospital admission from medical records and a survey questionnaire. At first, participants were briefly informed about the objectives of the survey. All participants completed the written informed consent before data collection. Then all required data were collected through face-to-face method by a trained interviewer at Community Healthcare Service. Ethical approval was given by Medical Ethic Committee of Shenzhen Center for Disease Control and Prevention. Individual consent forms were translated into Mandarin and consent was obtained for all volunteers face to face. All participants held the right to withdraw from the study at any stage.

### 2.2. Data Collection 

Data was obtained by trained personnel, who interviewed all the participants face to face. Participants would undergo a standardized physical examination, anthropometric measurements, blood routine and urine routine test, based on the prepared questionnaire. The following information was also collected: age, gender, weight (kilograms), height (centimeters), blood pressure, education level, dietary habits (dietary pattern, tea drinking, alcohol consumption and so on), dietary supplement such as medical usage (lipid-lowering agents, blood pressure medication, hypoglycemic agents, insulin, pain killer, anticoagulants, soporific, asthma medicine, antitussive, amcinonide and so on), antibioticusage (penicillin, floxacin, polymyxin, metronidazole, cephalosporins, streptomycin, tetracycline, erythromycin, carbopenems, glycopeptides and so on) and probiotic supplementation, life styles (smoking history, level of physical activity, degree of satiety, adoptpets or not) and other items. Body mass index (BMI) was calculated as body weight in kilograms divided by height in meters squared. Venous blood samples were collected after an overnight fast (at least 8 h) for measurement of biochemistry indicators. 

A qualitative food frequency questionnaire (Q-FFQ) was used to assess dietary data. The Q-FFQ covered the dietary frequency of 24 major food groups and consumption amount of 20 food groups. The following food groups were included: beef, poultry, pork, seafood, fish, dairy products, soybean products, nuts, eggs, yoghurt, fruits, vegetables, potatoes anddessert. Dietary factors were derived from the 20 food groups according to the Chinese balance dietary pagoda. 

### 2.3. Direct Sample Testing with PCR and NDM Genes Quantification with qPCR

Samples were enriched in 50 mL of BHI broth (Land Bridge, Beijing, China) containing 0.25 mg/L meropenem and 30 mg/L vancomycin at 37 °C overnight. Total DNA were extracted from 1 mL of each enriched culture described above via the boiling method and used for a PCR. PCR amplifications were performed using a Biometra thermocycler (Biometra T Gradient, Germany). The total DNA of the fecal samples were extracted via a DNeasePowerSoil Kit (QIAGEN, Hilden, Germany) according to the manufacture’s manuscript protocol. The concentration and quality of the extracted DNA samples were determined via a Nanodrop 2000 spectrophotometer (Thermo Fisher Scientific, Waltham, MA, USA). Primers NDM-F (5′-TGGCAGCACACTTCCTATCTC-3′) and NDM-R (5′-ATACCGCCCATCTTGTCCTG-3′) were used to quantify the abundance of *bla*_NDM_ in the samples by quantitative PCR (qPCR) via the ABI 7500 sequencer (Life Technologies, Inc., Carlsbad, CA, USA) with PowerUp SYBR Premix Ex Taq II (Thermo Fisher Scientific, Waltham, MA, USA). The optimized experimental protocol for qPCR was as follows: 95 °C for 2 min, 40 cycles at 95 °C for 10 s and 60 °C for 40 s. The data were analyzed with the manufacturer’s software (ABI 7500 software 2.3, Applied Biosystems, Foster, CA, USA). The target genes NDM were obtained by PCR amplification from the DNA extracts, and validated by sequencing (BGI, Beijing, China). The target genes were then cloned into *E.coli* DH5α using pMD19-T vector. The plasmid containing the target genes were extracted and used as positive controls in PCR and qPCR assays.

### 2.4. Statistical Analysis

The characteristics were expressed as the mean and percentages for continuous and categorical variables, respectively. Baseline demographic and laboratorial characteristics were presented as mean ± standard deviation (SD) or a number with percentage. The differences between the groups were examined by the Student’s *t*-test, or Chi-square test, depending on data types. Logistic regression models were conducted to assess the odds ratio (OR) and 95% confidence interval (95% CI) for each dietary factor on *bla*_NDM_ carrier. Specifically, an association model was developed to study the dietary intake that best predicts *bla*_NDM_ carrier. First, all categorized food groups were univariately tested in the model with absence or presence of *bla*_NDM_ gene as an outcome. All categorized food groups with *p* < 0.2 were included in the association model, which also included the previously described potential confounders: sex, age, number of previously prescribed antimicrobial drugs (penicillin, floxacin, polymyxin and metronidazole), diabetes, triglycerideindex, body mass index (BMI). Next, backward selection with *p* < 0.2 as end point (only for the food groups) was conducted to determine the final model. Spearman’s correlation coefficients for the food groups were calculated. To prevent collinearity, food groups with a correlation >0.7 were not added to the same model. The variance inflation factor (VIF) was calculated to evaluate the multicollinearity of multiple models. A VIF < 10 indicates that the multicollinearity may not affect the estimations [21]. The Hosmere-Lemeshow test was performed to test the goodness-of-fit of the model. Propensity score matching (PSM) was performed using psmatch2 (Figure 1). All statistical analyses were performed with IBM SPSS Statistics 24.0 (SPSS Inc., Chicago, IL, USA), STATA 12.0 (StataCorp, Texas City, TX, USA) and R 3.3.3 (Lucent Technologies, Murray Hill, NJ, USA). *p* < 0.05 was considered statistically significant.

## 3. Results

### 3.1. Laboratory Investigations

For the prevalence study, we analyzed 668 fecal samples from health communities living in Shenzhen, Guangdong, China. We found 99 *bla*_NDM_ positive samples through qPCR analysis. While only 35 positivesweredetected through direct sample testing and only 3 isolates resistantto 0.25 mg/L meropenem and 30 mg/L vancomycin at 37 °C were selected and cultured finally (data not shown).

### 3.2. Demographic Characters 

The general demographic characteristics of participants are summarized in Table 1. The mean age was 57 years. No significant difference between the *bla*_NDM_ carrier and *bla*_NDM_ non-carrier was observed with respect to age, gender, medical contact in recent three months, gastrointestinal disorders (including intestinal inflammation, peptic ulcer, irritable bowel syndrome, gastrointestinal bleeding), hypertension, hyperlipemia andsmoking status (*p* > 0.05). Compared to the non-carrier, participants carrying *bla*_NDM_ were reported greater prevalence with BMI, dietary factor (meat-based, vegetarian-based or balanced diet), degree of satiety, blood triglyceride and were more likely to have mental illness at the same moment (*p* < 0.05). Otherwise, the prevalence between *bla*_NDM_ carrier and antibiotic usage in recent three months (including penicillin, floxacin, polymyxin and metronidazole), drinking status, physical exercise and diabetes were reported marginally significant (*p* < 0.1). Other demographics were summarized in Appendix A.

### 3.3. Dietary Factor

Of all participants in this study, 515 had a fecal culture collection, total DNA extraction and completed an FFQ. The associations between FFQ and *bla*_NDM_ carrier are shown in Table 2. The *bla*_NDM_ carrier was significantly associated with greater intake of root and tuber crops and coarse grains.The *bla*_NDM_ carrier was significantly associated with less intake of poultry, fruit and yogurt (*p* < 0.05). Since participants carrying *bla*_NDM_ were reported greater prevalence of BMI*,* the *p* value was adjusted by BMI quartile. As shown in Table 2, the additional adjustment for different food intake did not substantially change the magnitude of these associations. All categorized food groups were univariately tested in the model with absence or presence of *bla*_NDM_ gene as an outcome (Table 3). Most FFQ were low correlated with each other, and the Spearman’s rank correlation coefficients ranged from −0.007 to 0.402 (Appendix A). A higher intake of root and tuber crops was associated with *bla*_NDM_ carrier (OR 1.001; 95% CI 1.000–1.003). Furthermore, a higher intake of coarse grains was associated with *bla*_NDM_ carrier (OR 1.002; 95% CI 1.001–1.004). In contrast, a recommended intake of fruit was associated with lower prevalence of *bla*_NDM_ carrier (OR 0.337; 95% CI 0.142–0.801) (Table 3).

### 3.4. Multivariable Logistic Regression Analysis 

Multivariable logistic regression analysis was then applied to explore the significance of these food factors. After selecting the food groups and demographics with a cut-off of *p* < 0.2 (see Table 1, Table 2 and Appendix A), an association model was made using backward selection. To prevent collinearity, food groups with a correlation >0.7 and VIF > 10 were not added to the same model. Most of the correlation coefficients between the different food groups as well as demographics were low and none of them was >0.7 (see Appendix A). VIF of different food groups were less than 2.0 (Figure 2). The results (model 1 as shown in Table 3 and Figure 2) revealed that drinkers (OR 2.348; 95% CI 1.241–4.442, *p* < 0.01) and coarse grains (OR 1.003; 95% CI 1.001–1.005, *p* < 0.01) were independent risk factor for *bla*_NDM_ carrier in health communities as shown in Table 3. While antibiotic usage in recent three months (OR 0.302; 95% CI 0.097–0.937, *p* < 0.05), physical exercise (OR 0.503; 95% CI 0.277–0.916, *p* < 0.05) and the intake of recommended amount of fruit (OR 0.310; 95% CI 0.121–0.795, *p* < 0.05) were independent protective factor for *bla*_NDM_ carrier in health communities as shown in Table 3 and model 1.

According to the results of multivariable analysis, we found that several factors would act as confounding factor. Then we applied a PSM by gender, age, BMI, triglyceride, FBS, physical exercise and pet. A ratio of 1:3 PSM was obtained, and a conditional multivariable logistic analysis was applied to explore the association between *bla*_NDM_ carrier and dietary factors such as drinkers, antibiotic usage in recent three months, degree of satiety, dietary pattern as well as FFQ. Additionally, drinkers, antibiotic usage in recent three months, degree of satiety anddietary pattern were omitted with the model (model 2 as shown in Table 3) because of no within-group variance. The results conditional logistic analysis after PSM revealed of that the intake of coarse grain (OR 1.003; 95% CI 1.001–1.005, *p* < 0.01) and root and tuber crops (OR 1.003; 95% CI 1.001–1.004, *p* < 0.05) were independent risk factor for *bla*_NDM_ carrier in health communities as shown in Table 3. While poultry intake (OR 0.996; 95% CI 0.992–1.000, *p* < 0.05) and the intake of recommended amount of fruit (OR 0.208; 95% CI 0.081–0.539, *p* < 0.01) were independent protective factor for *bla*_NDM_ carrier in health communities as shown in Table 3 and model 2.

## 4. Discussion

Anti-Microbial Resistance (AMR) is a pandemic which threatens modern medicine. Antimicrobial ingredients in antibacterial hand sanitizers, hand-wash lotions and chlorine-containing disinfectants would promote genetic mutations in bacteria that lead to antibiotic resistance and accelerate the spread of resistance genes between bacteria. The overuse of these antimicrobial products during the COVID-19 pandemic may speed up these procedures and thereby expedite the onset of a superbug pandemic that seriously threatens future public health [22]. The emergence and spread of CRE carrier and infections are serious threats to public health worldwide. Carbapenem-resistant gene *bla*_NDM_ was frequently found from non-human sources [8,9,10,11,12,13,14,15]. There is an ongoing debate as to what extent antimicrobial resistance (AMR) can be transmitted from animals to humans via the consumption of animal products. In order to explore this topic, the present study aspired to investigate the association between *bla*_NDM_ carrier and anthropometric, biochemical and dietary factors in health community participants living in Shenzhen, China.

In terms of the prevalence of *bla*_NDM_ carrier among health communities, 99, 35 and 3 *bla*_NDM_ positive were found respectively via qPCR analysis, direct sample testing and selected isolation. Several reasons may contribute to these differences. First, gene *bla*_NDM_ is not necessarily present in living bacteria, but may be present in the environment which might degrade rapidly before detection. Second, we mainly screened Enterobacteriaceae bacteria. These bacteria may not contain the target gene and the bacteria containing *bla*_NDM_ gene was omitted by our selecting culture process. Third, the resistance ability between bacteria isolated from health community and hospital was significantly different which might escape the selecting culture process [17].

It has already been reported that medical contacts, includingprior antibiotic use, prior hospital stay, exposure to healthcare facilities, biliary drainage catheter, vascular device, tracheostomy, transferring to intensive care unit (ICU), abdominal invasive procedure and so on, were risk factors for patients to have CRE [23,24,25,26,27]. We also showed an association between a high intake of vegetables especially root and tuber crops with*bla*_NDM_ carrier. This confirms the results of our previous study, which showed an association between a vegetables-based dietwith carbapenem resistance.For poultry, it is hypothesized that contaminated poultry can possibly transfer resistant pathogens [28]. The association between a high intake of chicken and resistome colonization in health community participants suggests a possible transmission of gene *bla*_NDM_ via food. Even though alarming, we could not find any direct evidence that this results in carrier of gene *bla*_NDM_ and the intake of poultry. Thus, the possibility that poultry products play a direct role in the transmission of antimicrobial resistance from animals to humans is not very likely according to the results of this study.

Certain food groups are able to affect the human gut microbiota [16,29]. The resistome and dysbiosis of the gut microbiota were considered to speed upthe horizontal gene transfer of resistance genes in the gut [30,31]. As we all know, the dietary intervention could diminish the gut resistome [32]. Additionally, a potential beneficial effect of diets rich in vegetables and cheese on the gut microbiota has been described [33,34]. These studies were consistent with our results vegetarian turned to get less *bla*_NDM_ carrier (Table 1). 

As shown in Table 1 and Appendix A, physical exercise and BMI were associated with *bla*_NDM_ carrier. Specifically, less physical activity and overweight status were associated with higher *bla*_NDM_ carrier. Individuals with a less healthy lifestyle were more likely to have less favorable dietary patterns and this mightlead to a higher risk of *bla*_NDM_ carrierand obesity.

Furthermore, polysaccharides and fiberswhich consist of the main potatoes [35], have been proposed to benefit the gut microbiome [36]. While in our result, an association between a high intake of vegetables especially root and tuber crops (including potatoes) and *bla*_NDM_ carrier were found. These contrary results need more evidence to uncover the mechanism how dietary pattern influence gut microbiome and AMR colonization.

In conclusion, we showed that a high intake of coarse grain (OR 1.003; 95% CI 1.001–1.005, *p* < 0.01) and root and tuber crops (OR 1.003; 95% CI 1.001–1.004, *p* < 0.05) were independent risk factor for *bla*_NDM_ carrier in health communities, suggesting a possible transfer of antimicrobial resistance from food animals to humans. Surprisingly, we also showed an association between a dietary pattern as vegetarian, which may be explained by the beneficial effects on the gut microbiota. 

### 4.1. Strengths and Limitation

The strengths of thisstudy included results that would be useful for health professionals to identify the high-risk factors of the fecal carrier of *bla*_NDM_ gene. This information would benefit the health authorities and policymakers to develop specific screening guidelines based on dietary indicators that would best identify high-risk individuals of *bla*_NDM_ carrier in the community for appropriate intervention. As for the implicationsof the findings for policy and practice, the influence of dietary factors as well as demographic, anthropometric, lifestyle and biochemical characteristicson *bla*_NDM_ carrier in the present study also provides insights for the tangible dietary advice, reference lifestyles and primary health care with guidelines to the routine of people with the risk of *bla*_NDM_ carrier.

However, our study had limitations. First, this is a cross-sectional study, and we cannot draw conclusions about causeandeffect between dietary patterns and resistome carriers in health communities. Second, the results may not be generalized to the general population. Becauseour participants also came from that of a specific region of China and, therefore, may not be representative of the Chinese’s population. Third, self-report of dietary intake by questionnaire used in this study, constitutes a well-known weakness due to recall bias and social desirability bias. This commonly used method due to its convenience and low-cost is considered to be a crude method andmeans that there is an obvious risk of misclassification of the number of portions consumed. 

### 4.2. Future Research

Generally, our results need more evidence to uncover the mechanism how dietary factors influence gut microbiome and AMR colonization. It is worth to verify whether changes in dietary intake have any effect on *bla*_NDM_ gene carrier as well as the ‘resistome’ and dysbiosis of the gut microbiota. Future research should also aim at developing and evaluating interventions how the increasing intake of vegetable improve the capture of *bla*_NDM_ gene in the population, and especially among the overweight population.

Since the protective role of dietary factor in the capture of *bla*_NDM_ gene was confirmed in this study, the associations between biochemical parameters and food consumption were also highlighted and drew our attention. Thus, the role of the relationship between food consumption and biochemical parameters in the impact of *bla*_NDM_ gene carrier is still in question and needs more research. Furthermore, it seems important to investigate if specific types of food are more protective than others by investigating mechanisms of action. To do that, other food intake surveys such as a food diary and a better monitoring of nutritional biomarkers of intakesuch as ideally blood samples would be required.

## 5. Conclusions

In conclusion, the results of our study suggest that *bla*_NDM_ carrier were associates with coarse grain (OR 1.003; 95% CI 1.001–1.005, *p* <0.01) and root and tuber crops(OR 1.003; 95% CI 1.001–1.004, *p* <0.05). In order to clarify the mechanisms behind these differential effects of dietary FFQ, more research is needed with a more detailed assessment of the impacts of food intake for this population.

The influence of dietary factors on *bla*_NDM_ carrier in the present study provides insights for the tangible dietary advice with guidelines to the routine of people with the risk of *bla*_NDM_ carrier. This demonstrates the role of dietary intake in the prevention of *bla*_NDM_ carrier, since prevention is the best way to control modifiable risk factors. A lower carrier rate of *bla*_NDM_ is helpful to reduce the possibility of transmission and pathogenicity. Further studies on food, microbiota and antimicrobial resistance are necessary to confirm this possible association and unravel underlying mechanisms.

## Figures and Tables

**Figure 1 ijerph-18-05959-f001:**
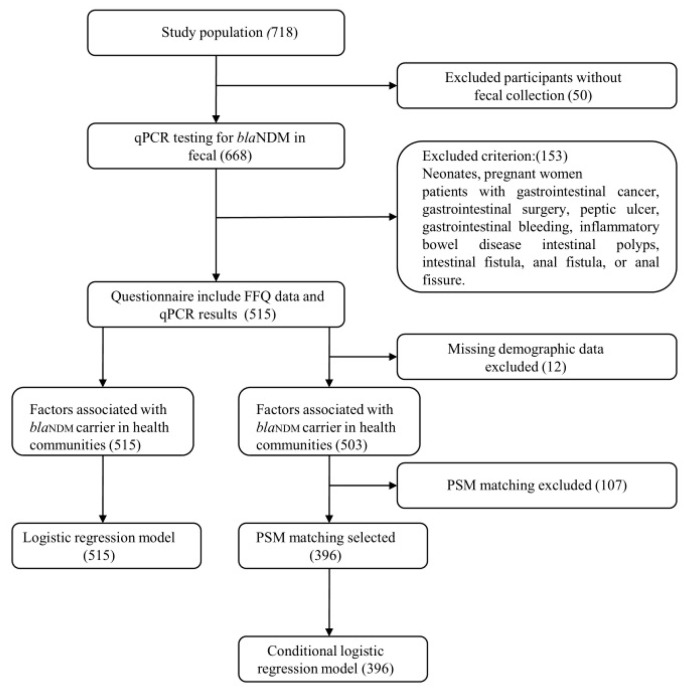
Flowchart of the study.

**Figure 2 ijerph-18-05959-f002:**
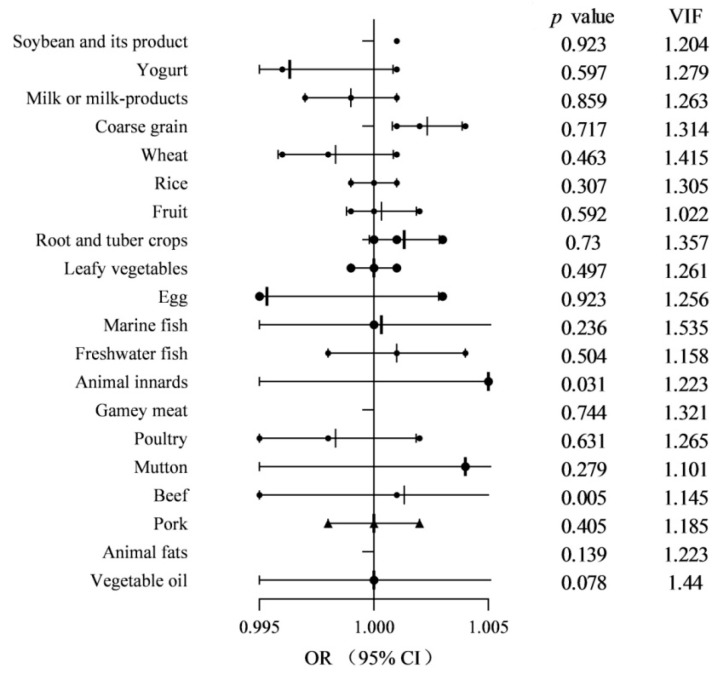
The multiple FFQ model for the associations of FFQ with the risk of *bla*_NDM_ carrier.OR, odds ratio; VIF, variance inflation factor; 95% CI, 95%confidence interval. The dotsrepresent crude oddsratios, and the horizon lines represent 95% confidence intervals. *p*-valueswere obtained from the univariate logistic regression models by using the FFQ as a continuous variable.

**Table 1 ijerph-18-05959-t001:** Baseline demographics of the participants with or without fecal carriage of *bla*_NDM_ gene.

Variables	Rank	All Participants (*n* = 515)	*bla*_NDM_ Carrier (*n* = 99)	*bla*_NDM_ Non-Carrier (*n* = 416)	*bla*_NDM_ Carrier vs. Non-Carrier, *p*-Value
Age (years)		57.16 ± 12.157	57.19 ± 12.193	57.02 ± 12.063	0.953
BMI		24.26 ± 3.146	25.51 ± 7.868	24.27 ± 3.448	0.038 *
Gender, n (%)	Male	180 (35.0)	39 (39.4)	141 (33.9)	0.302
	Female	335 (65.0)	60 (60.6)	275 (66.1)	
Medical contact in recent three months, n (%)	No	31 (6.0)	6 (6.1)	25 (6.0)	0.985
	Yes	484 (94.0)	93 (93.9)	391 (94.0)	
Antibiotic usage in recent three months, n (%)	No	466 (91.4)	94 (95.9)	372 (90.3)	0.075
	Yes	44 (8.6)	4 (4.1)	40 (9.7)	
Smoking, n (%)	No	421 (81.7)	82 (82.8)	339 (81.5)	0.725
	Yes	42 (8.2)	9 (9.1)	33 (7.9)	
	Quit	52 (10.1)	8 (8.1)	44 (10.6)	
Drinking, n (%)	No	411 (79.8)	76 (76.8)	335 (80.5)	0.078
	Yes	73 (14.2)	20 (20.2)	53 (12.7)	
	Quit	31 (6.0)	3 (3.0)	28 (6.7)	
Physical exercise, n (%)	No	89 (17.3)	23 (23.2)	66 (15.9)	0.081
	Yes	426 (82.7)	76 (76.8)	350 (84.1)	
Dietary pattern, n (%)	Meat-based	31 (6.0)	4 (4.0)	27 (6.5)	0.008 **
	Balanced diet	372 (72.2)	66 (66.7)	306 (73.6)	
	Vegetarian-based	110 (21.4)	27 (27.3)	83 (20.0)	
	Not clear	2 (0.4)	2 (2.0)	0 (0.0)	
Degree of satiety, n (%)	Full	45 (8.8)	7 (7.1)	38 (9.2)	0.046 *
	90 percent	165 (32.1)	21 (21.2)	144 (34.7)	
	80 percent	218 (42.4)	56 (56.6)	162 (39.0)	
	70 percent	78 (15.2)	14 (14.1)	64 (15.4)	
	60 percent	6 (1.2)	1 (1.0)	5 (1.2)	
	<60 percent	2 (0.4)	0 (0.0)	2 (0.5)	
Pet	No	482 (94.0)	96 (97.0)	386 (93.2)	0.161
	Yes	31 (6.0)	3 (3.0)	28 (6.8)	
Hypertension, n (%)	No	388 (75.3)	71 (71.7)	317 (76.2)	0.352
	Yes	127 (24.7)	28 (28.3)	99 (23.8)	
Hyperlipemia, n (%)	No	411 (80.0)	82 (82.8)	329 (79.3)	0.428
	Yes	103 (20.0)	17 (17.2)	86 (20.7)	
Obesity, n (%)	No	467 (90.7)	91 (91.9)	376 (90.4)	0.637
	Yes	48 (9.3)	8 (8.1)	40 (9.6)	
Diabetes, n (%)	No	472 (91.7)	95 (96.0)	377 (90.6)	0.085
	Yes	43 (8.3)	4 (4.0)	39 (9.4)	
Gastrointestinal disorders, n (%)	No	435 (84.5)	80 (80.8)	355 (85.3)	0.264
	Yes	80 (15.5)	19 (19.2)	61 (14.7)	
Mental illness, n (%)	No	511 (99.4)	97 (98.0)	414 (99.8)	0.037 *
	Yes	3 (0.6)	2 (2.0)	1 (0.2)	
Blood urine acid	<202 μmol/L (male) <142 μmol/L (female)	6 (1.2)	3 (3.0)	3 (0.7)	0.150
	202~416 μmol/L (male) 142~339 μmol/L (female)	351 (68.2)	65 (65.7)	286 (68.8)	
	>416 μmol/L (male) >339 μmol/L (female)	158 (30.7)	31 (31.3)	127 (30.5)	
Triglyceride, n (%)	<1.7 mmol/L	346 (67.2)	58 (58.6)	288 (69.2)	0.043 *
	≥1.7 mmol/L	169 (32.8)	41 (41.4)	128 (30.8)	
Fast blood sugar (FBS), n (%)	<3.9 mmol/L	6 (1.2)	3 (3.0)	3 (0.7)	0.109
	3.9~6.1 mmol/L	443 (86.0)	81 (81.8)	362 (87.0)	
	>6.1 mmol/L	66 (12.8)	15 (15.2)	51 (12.3)	

ANOVA was used for statistical comparisons between quantitative variables, and chi-square test was used for statistical comparisons between qualitative variables.Weighted values are means and theirSD (continuous variables) or percentages (categorical variables).* *p* < 0.05, ** *p* < 0.01 significantly associated with prevalence of *bla*_NDM_ carrier.

**Table 2 ijerph-18-05959-t002:** Association between dietary intake of the participants and the carriage of *bla*_NDM_ gene.

Figure 515.	Rank	All Participants (*n* = 515)	*bla*_NDM_ Carrier (*n* = 99)	*bla*_NDM_ Non-Carrier (*n* = 416)	*bla*_NDM_ Carrier vs. Non-Carrier, *p*-Value	Adjust *p*-Value
Oil	Lack	111 (21.6)	16 (16.2)	95 (22.8)	0.149	0.145
	Recommended	82 (15.9)	21 (21.2)	61 (14.7)		
	Over	322 (62.5)	62 (62.6)	260 (62.5)		
Vegetable oil		49.70 ± 35.66	49.39 ± 29.07	49.77 ± 37.09	0.232	
Animal fats		2.60 ± 8.84	3.02 ± 9.57	2.50 ± 8.67	0.294	
Livestock and poultry	Lack	220 (42.7)	44 (44.4)	176 (42.3)	0.892	0.897
	Recommended	182 (35.3)	33 (33.3)	149 (35.8)		
	Over	113 (21.9)	22 (22.2)	91 (21.9)		
Pork		87.84 ± 99.25	89.43 ± 98.27	87.46 ± 99.59	0.957	
Beef		16.15 ± 31.27	17.18 ± 32.93	15.91 ± 30.90	0.800	
Mutton		4.22 ± 17.87	5.44 ± 25.34	3.93 ± 15.60	0.350	
Poultry		41.49 ± 77.00	34.32 ± 58.82	43.19 ± 80.69	0.043 *	
Gamey meat		0.50 ± 4.71	0.25 ± 1.71	0.56 ± 5.17	0.359	
Animal innards		2.13 ± 7.30	2.36 ± 6.30	2.08 ± 7.53	0.830	
Aquatic product	Lack	306 (59.4)	58 (58.6)	248 (59.6)	0.516	0.524
	Recommended	98 (19.0)	16 (16.2)	82 (19.7)		
	Over	111 (21.6)	25 (25.3)	86 (16.7)		
Freshwater fish		43.48 ± 75.88	48.15 ± 76.55	42.36 ± 75.77	0.771	
Marine fish		15.92 ± 32.24	16.20 ± 28.26	15.85 ± 33.15	0.731	
Egg	Lack	202 (39.2)	41 (41.4)	161 (38.7)	0.517	0.508
	Recommended	273 (53.0)	53 (53.5)	220 (52.9)		
	Over	40 (7.8)	5 (5.1)	35 (8.4)	0.030 *	0.030 *
Vegetables	Lack	196 (38.1)	28 (28.3)	168 (40.4)		
	Recommended	206 (40.0)	41 (41.4)	165 (39.7)		
	Over	113 (21.9)	30 (30.3)	83 (20.0)		
Leafy vegetables		256.62 ± 190.15	268.14 ± 148.09	253.88 ± 198.91	0.293	
Root and tuber crops		127.18 ± 148.89	156.96 ± 207.11	120.09 ± 130.63	0.002 **	
Fruit	Lack	438 (85.0)	92 (92.9)	346 (83.2)	0.036 *	0.033 *
	Recommended	73 (14.2)	6 (5.1)	67 (16.1)		
	Over	4 (0.8)	1 (1.0)	3 (0.7)		
Grains	Lack	12 (2.3)	1 (1.0)	11 (2.6)	0.135	0.155
	Recommended	70 (13.6)	19 (19.2)	51 (12.3)		
	Over	433 (84.1)	79 (79.8)	354 (85.1)		
Rice		305.71 ± 255.39	294.62 ± 247.09	308.35 ± 257.55	0.461	
Wheat		77.21 ± 86.80	68.70 ± 93.17	79.24 ± 85.21	0.311	
Coarse grains		89.27 ± 114.69	119.38 ± 159.80	82.11 ± 99.97	0.000 *	
Dairy product	Lack	456 (88.5)	93 (93.6)	363 (87.3)	0.061	0.057
	Recommended	0 (0.0)	0 (0.0)	0 (0.0)		
	Over	59 (11.5)	6 (6.1)	53 (12.7)		
Milk or milk-products		94.38 ± 118.49	85.47 ± 104.56	96.51 ± 121.59	0.129	
Yogurt		29.66 ± 62.64	21.19 ± 41.33	31.67 ± 66.60	0.010 *	
Soybean and its product	Lack	408 (79.2)	85 (85.9)	323 (77.6)	0.194	0.196
	Recommended	22 (4.3)	3 (3.0)	19 (4.6)		
	Over	85 (16.5)	11 (11.1)	74 (17.8)		

ANOVA was used for statistical comparisons between quantitative variables, and chi-square test was used for statistical comparisons between qualitative variables.Values shown are based on mean ± SD. Adjust *p* value, *p* value adjusted by BMI quartile. * *p* < 0.05, ** *p* < 0.01 significantly associated with prevalence of *bla*_NDM_ carrier.

**Table 3 ijerph-18-05959-t003:** Odds ratios (95% confidence intervals) forfecal carriage of*bla*_NDM_ gene associated with dietary FFQ.

Food Groups		Crude					Model 1			Model 2			
		OR		95% CI	*p*	OR		95% CI	*p*	OR	95% CI		*p*
Drinking	No	1.000			0.085	1.000			0.006 **				
	Yes	1.663	0.939	2.946	0.081	2.348	1.241	4.442	0.009 **				
	Quit	0.472	0.140	1.594	0.227	0.264	0.056	1.247	0.093				
Antibiotic usagein recent three months	No	1.000				1.000							
	Yes	0.396	0.138	1134	0.084	0.302	0.097	0.937	0.038 *				
Pet	No	1.000				1.000							
	Yes	0.431	0.128	1.447	0.173	0.353	0.099	1.263	0.109				
Physical exercise	No	1.000				1.000							
	Yes	0.623	0.365	1.064	0.083	0.503	0.277	0.916	0.025 *				
Degree of satiety, n (%)	Full	1.000			0.062	1.000			0.059				
	90 percent	0.792	0.313	2.001	0.621	0.749	0.269	2.085	0.580				
	80 percent	1.877	0.793	4.441	0.152	1.895	0.725	4.955	0.192				
	70 percent	1.188	0.440	3.203	0.734	1.058	0.351	3.188	0.920				
	60 percent	1.086	0.110	10.758	0.944	2.468	0.217	28.097	0.467				
	<60 percent	<0.001	0.000		0.999	<0.001	<0.001		1.000				
Dietary pattern, n (%)	Meat-based	1.000			0.349	1.000			0.451				
	Balanced diet	1.456	0.493	4.301	0.497	1.504	0.466	4.852	0.494				
	Vegetarian-based	2.196	0.705	6.840	0.175	2.238	0.647	7.735	0.203				
	Not clear	<0.001	0.000		0.999	>10	<0.001		0.999				
Poultry		0.998	0.995	1.002	0.307	0.997	0.993	1.001	0.128	0.996	0.992	1.000	0.037 *
Root and tuber crops		1.001	1.000	1.003	0.031 *	1.002	1.000	1.003	0.051	1.003	1.001	1.004	0.002 **
Fruit	Lack	1.000			0.047 *	1.000			0.048 *	1.000			
	Recommended	0.337	0.142	0.801	0.014 *	0.310	0.121	0.795	0.015 **	0.208	0.081	0.539	0.001 **
	Over	1.254	0.129	12.193	0.846	1.483	0.129	17.113	0.752	0.892	0.000	-	0.988
Coarse grains		1.002	1.001	1.004	0.005 **	1.003	1.001	1.005	0.001 **	1.003	1.001	1.005	0.005 **
Yogurt		0.996	0.992	1.001	0.139	0.996	0.991	1.001	0.139	0.995	0.989	1.000	0.060

OR, odds ratio; CI, confidence intervals.Model 1 adjusted for none. Model 2 adjusted for all covariates *p*< 0.2 listed in Table 1. * *p* < 0.05, ** *p* < 0.01 significantly associated with prevalence of *bla*_NDM_ carrier.

## Data Availability

The data presented in this study are available on request from the corresponding author. The data are not publicly available due to personal data protection.

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
