# Peer review of "Dietary Factors of blaNDM Carriage in Health Community Population: A Cross-Sectional Study"

_ijerph, 2021, doi:10.3390/ijerph18115959_

Round 1

Reviewer 1 Report

This manuscript seems about ready for publication.

It will need thorough editing to achieve journal-level language proficiency.

One minor quibble is that it would be preferable to refer to "multivariable" rather than "multivariate" logistic regression.

The authors are encouraged to add some discussion about implications of the findings for policy and practice.

Author Response

Response to Reviewer 1 Comments

This manuscript seems about ready for publication.

Point 1: It will need thorough editing to achieve journal-level language proficiency.

Response 1:Thanks for your valuable comments. To meet the journal-level language proficiency, the manuscript has beenpolished thoroughly by a professional English editing service.

Point 2:One minor quibble is that it would be preferable to refer to "multivariable" rather than "multivariate" logistic regression.

Response 2:Thanks for your valuable suggestion. We have replaced " multivariate " with " multivariable " throughout the manuscript.

Point 3:The authors are encouraged to add some discussion about implications of the findings for policy and practice.

Response 3:Thanks for your valuable suggestion. We have added context about implicationsof the findings for policy and practice in the discussion section.As for the implicationsof the findings for policy and practice,the influence of dietary factors as well as demographic, anthropometric, lifestyle and biochemical characteristics on blaNDM carrier in the present study also provides insights for the tangible dietary advice, reference life styles and primary health care with guidelines to the routine of people with the risk of blaNDM carrier.

Reviewer 2 Report

Revision of manuscript: Dietary factors of blaNDM carriage in health community population: a cross-sectional study

This is an interesting cross-sectional study about dietary factors of blaNDM carriage in health community population: a cross-sectional study. Most of the methods of the study are well-described, and most of the inclusion and exclusion criteria clearly stated. Also, the statistical analyses are precisely explained. The topic is important and most of the issues are discussed and explained in the discussion section.

However, there are some serious limitations in the manuscript, which should be improve before publication in International Journal of Environmental Research and Public Health.

First of all, in the inclusion criteria there is lack of information about other medicaments (is only a question about antibiotics) and no information about probiotic supplementation. I think that authors should complete this information.

The aim of the study should be standardized. The aim in the abstract  is different than in the introduction section. Once we have dietary intake, once we have dietary patterns. In the title we have dietary factors.

These phrases are not synonyms. Authors have to improve those mistakes. It has to be added that the study presented in this manuscript is not only about dietary intake (as the aim of study indicated). In the table authors added many information about life style for example: level of physical activity, degree of satiety, having a pets. Besides, there are many information about health state of participants. That why a think that the title of this manuscript  and the aim in the abstract and in the introduction should be reformulated.

Author Response

Response to Reviewer 1 Comments

Revision of manuscript: Dietary factors of blaNDM carriage in health community population: a cross-sectional study This is an interesting cross-sectional study about dietary factors of blaNDM carriage in health community population: a cross-sectional study. Most of the methods of the study are well-described, and most of the inclusion and exclusion criteria clearly stated. Also, the statistical analyses are precisely explained. The topic is important and most of the issues are discussed and explained in the discussion section.However, there are some serious limitations in the manuscript, which should be improve before publication in International Journal of Environmental Research and Public Health.

Point 1:First of all, in the inclusion criteria there is lack of information about other medicaments (is only a question about antibiotics) and no information about probiotic supplementation. I think that authors should complete this information.

Response 1:Thanks for your valuable comments. In fact, dietary supplement such as medical usage (lipid-lowering agents, blood pressure medication, hypoglycemic agents, insulin, pain killer, anticoagulants, soporific, asthma medicine, antitussive, amcinonide and so on), antibiotic usage (penicillin,floxacin,polymyxin,metronidazole,cephalosporins,streptomycin,tetracycline,erythromycin,carbopenems,glycopeptides and so on) and probiotic supplementationhave also been surveyed with the questionnaire. While according to theunivariatetestin the model with absence or presence of blaNDM gene as an outcome, most of these factors have no association with blaNDM carrier.Therefore, these results werebrief and uninformative in the manuscript.

Point 2:The aim of the study should be standardized. The aim in the abstract is different than in the introduction section. Once we have dietary intake, once we have dietary patterns. In the title we have dietary factors. These phrases are not synonyms. Authors have to improve those mistakes.

Response 2:Thanks for your valuable comments. Dietary intake defined as the intake amount of a particular kind of food or drink. In this manuscript, dietary intake of particular kind of food were obtainedquestionnaire.Dietary factors indicated the conditions of people’s dietary intake. In this manuscript, dietary factorsindicated the qualitative food frequency questionnaire (Q-FFQ) assessing the dietary intake.Dietary pattern defined as the quantity, variety, or combination of different foods and beverage in a diet and the frequency with which they are habitually consumed.In this manuscript,dietary pattern indicated four kinds of dietary habits such as meat-based diet, balanced diet, vegetarian-baseddiet and not specific diet.We have revised the manuscript and make the usage consistent throughout the manuscript.

Point 3:It has to be added that the study presented in this manuscript is not only about dietary intake (as the aim of study indicated). In the table authors added many information about life style for example: level of physical activity, degree of satiety, having a pets. Besides, there are many information about health state of participants. That why a think that the title of this manuscriptand the aim in the abstract and in the introduction should be reformulated.

Response 3:Thanks for your valuable comments.In this study, life styles(such as level of physical activity, degree of satiety, having a pets), that might influence the AMR carriage in human and act as cofounders in the analysis, were obtained withquestionnaire. According to the results of a robust multivariable analysis, we found that several factors would act as confounding factor. Subsequently, to explore the association between dietary factors and blaNDM carrier, we applied a Propensity Score Matching(PSM) by gender, age, health state information (BMI, triglyceride, FBS), physical exercise and life style (adopt pet or not). A ratio of 1:3 PSM was obtained and a conditional multivariable logistic analysis was applied to explore the association between blaNDM carrier and dietary factors such as drinkers, antibiotic usage in recent three months, degree of satiety, dietary pattern as well as FFQ. To summary, the primary aim of this paper is to investigate the association between dietary factor and andblaNDMcarrier, using data from a health community population in Shenzhen City, Guangdong Province, China, through Feb. 1, 2018 to Dec. 31, 2019. During this procedure, life styles, health state and physical exercise that might influence the AMR carriage in human and act as cofounders in the analysis, were obtained with questionnaire.